# Self-Management Behaviours in Type 2 Diabetes Across Gulf Cooperation Council Countries: An Updated Narrative Review to Enhance Patient Care

**DOI:** 10.3390/healthcare13172247

**Published:** 2025-09-08

**Authors:** Ashokkumar Thirunavukkarasu, Aseel Awad Alsaidan

**Affiliations:** Department of Family and Community Medicine, College of Medicine, Jouf University, Sakaka 72388, Saudi Arabia; aaalsaidan@ju.edu.sa

**Keywords:** self-management, T2DM patients, digital health, personalized care, medication adherence

## Abstract

**Background and Objectives**: Type 2 diabetes mellitus (T2DM) remains a significant public health problem across Gulf Cooperation Council (GCC) nations because of advancements in urbanization alongside behavioural lifestyle changes and genetic predispositions. Specific self-management methods are fundamental in T2DM management because they provide better glycaemic control and decrease complications. Achieving a synthesis of updated evidence about self-management strategies and patient perception within GCC nations represents the primary objective of this narrative review. **Materials and Methods**: The studies included in the present review were retrieved from the Web of Science, Scopus, Medline, Saudi Digital Library, and Embase. We included peer-reviewed studies that were published from January 2020 to March 2025. The selected studies measured the self-management practices of adult T2DM patients by examining medication adherence, dietary patterns, blood glucose monitoring, and treatment barriers. **Results**: Research data indicate that patients demonstrate different levels of self-care management behaviours, where medication compliance is fair, but dietary patterns and physical activities remain areas of concern. High levels of knowledge deficits, cultural elements, and economic background substantially impact patients’ self-management practices. Patients indicate their need for enhanced and personalized care, better connections with healthcare providers, and interventions that consider their cultural backgrounds. **Conclusions**: Patients throughout the GCC region encounter ongoing difficulties that prevent them from performing their best at self-management, even though advanced healthcare facilities exist in this region. Therefore, it is critical to develop culturally sensitive patient-centered care, individualized educational programs, and adopt supportive digital solutions to enhance diabetes-related self-care management.

## 1. Introduction

Type 2 diabetes mellitus (T2DM) is a rapidly expanding public health problem that ultimately contributes to the majority of diabetes cases globally [1]. According to the International Diabetes Federation (IDF), currently, 537 million adults worldwide have diabetes; this number is expected to reach 853 million in 2050 [2]. The Middle East and North Africa (MENA) region, including the Gulf Cooperation Council (GCC) countries—Bahrain, Kuwait, Oman, Qatar, Saudi Arabia, and the United Arab Emirates—is burdened greatly, with prevalence rates being one of the highest in the world. This alarming trend is the result of rapid urbanization, sedentary lifestyles, dietary transitions, and genetic predispositions, and places an enormous burden on the healthcare system and economic resources [3,4,5].

Within this context, effective self-management of T2DM has emerged as a cornerstone of chronic disease control. Active patient participation in daily self-management of a condition, such as medication adherence, maintaining a healthy diet, engaging in regular physical activity, blood glucose self-monitoring, and follow-up appointments. Strong evidence suggests that structured self-management practices lead to better glycaemic control, decrease the risk of complications, improve quality of life, and overall lead to better patient outcomes [6,7,8,9]. Diabetes self-management is a dynamic, multifaceted process that requires more than just individual willpower [10,11,12]. Understanding these factors might lead to better identification of effective interventions that improve patient care and outcomes. Patients’ self-care practices in GCC nations are influenced by various cultural views about chronic diseases and family participation in care, Islamic fasting traditions, and local herbal medicinal beliefs [13,14,15]. Self-management behaviour promotion programs for T2DM patients must blend both the cultural norms and healthcare structures of GCC countries.

The need for self-management awareness has increased, but various populations perform it differently. For example, Habibi Soola A. et al. stated that predictors for self-management can be grouped into four levels: the individual (e.g., gender and education), interpersonal, organizational, and community levels [16]. A qualitative exploration by Peng Xi et al. stated that these behaviours are complex phenomena [17]. In another study, Mikhael EM et al. stated that insufficient self-care practices were observed among their participants [18]. Some studies have been conducted in the past to explore self-care practices among T2DM patients. However, the GCC region lacks completely synthesized evidence connecting current and updated patient practices with their perspectives. Previous reviews of diabetes self-management focused on either global patterns or medical treatments without adequately examining how patients from GCC settings perceive their barriers to care and how these factors shape their experiences. Therefore, an updated, comprehensive review becomes essential at this time because diabetes care continues to advance through digital tool adoption and mobile health app use, as well as structured educational initiatives. Hence, we performed this review to synthesize recent peer-reviewed evidence on self-management behaviours among patients with T2DM across GCC countries. Furthermore, we examined the barriers, facilitators, and key sociodemographic factors influencing diabetes self-management.

## 2. Search Strategies

The studies included in the present review were retrieved from Web of Science, Scopus, Medline, Saudi Digital Library, and Embase. These databases provided a range of studies related to the self-management behaviours of T2DM patients, especially in the GCC region. A summary of the search strategies is provided in Table 1. During the literature search, we used the Boolean operators (“AND”, “OR”, and “NOT”) to refine the search via keywords. The keywords were related to three major terms: T2DM, self-management behaviour, and contextual factors (barriers, motivators, and demographic factors), for example, (“type 2 diabetes” OR T2DM) AND (“self-management” OR “self-care”) AND (barriers OR facilitators) AND (Saudi Arabia OR Qatar OR UAE OR Kuwait OR Oman OR Bahrain). We included studies that focused on adult (18 years and above) T2DM patients, were peer-reviewed, and had a DOI number. Furthermore, only studies published in English between January 2020 and March 2025 were included. We limited our review to studies published between January 2020 and March 2025 to ensure that the synthesis reflects the most recent evidence on self-management practices in the GCC region. This timeframe was selected to obtain the updated evidence because healthcare delivery models, patient behaviours, and self-management strategies underwent notable changes following the COVID-19 pandemic. We excluded studies solely on other forms of diabetes (such as type 1 diabetes, gestational diabetes, etc.), grey literature, unpublished works, and studies where the self-care of T2DM patients was not a primary focus.

After applying these pre-defined criteria, we included 88 articles in the main text for results and discussion. Firstly, two independent reviewers reviewed the articles, and it was included after reviewing its title, abstract, and main text. In the event of a discrepancy between the first two reviewers, we asked the third reviewer to provide their input. The study selection process is described as a flow diagram in Figure 1.

## 3. Main Findings (Results)

### 3.1. Patient Practices in Self-Management

#### 3.1.1. Medication Adherence Behaviours

Adhering to the physician’s recommended dose, the schedule of antidiabetic medications is one of the key components for better glycaemic control. The main reasons patients do not adhere to their treatments are forgetfulness, adverse side effects, complicated dosage schedules, worries about insulin usage, and perceived medication ineffectiveness [19,20]. A study in Saudi Arabia by Alfulayw MR et al. demonstrated that more than one-third of T2DM patients were not compliant with the prescribed drugs. They stated that a suboptimal level of medication adherence was observed despite the high-quality and accessible healthcare services available in their area [21]. However, a recent study in Ajman, UAE, stated that about 90% of their study participants adhered to the medications [22]. Using the Diabetes Self-Management Questionnaire, Al Ubaidi et al. in Bahrain demonstrated that about 74% of T2DM patients had an optimal medication adherence score [23].

A cross-sectional survey by Nair SC et al. in the UAE reported that a sizable proportion of their participants had inadequate levels of health literacy [24]. People without comprehensive health insurance adherence are significantly lower than those with insurance, as stated by Allaham KK et al. [25]. Some studies of healthcare systems in GCC countries noted that expatriates often face challenges in accessing medications because of high out-of-pocket costs, leading to treatment disturbance and nonadherence [25,26]. Some patients delay starting insulin treatment because of local traditions and social media influence related to this therapy, even though their medical situation requires it [27,28].

#### 3.1.2. Dietary Pattern Trends

Diabetic patients within the GCC region face dietary regulations as their main obstacle in regard to diabetes self-management. Many GCC countries maintain traditional diets containing multiple carbohydrates, saturated fats, and sugars, and they practice solid cultural customs to show hospitality through communal meals featuring high-calorie foods [29,30,31]. A study by Al-Mssallem MQ from Saudi Arabia demonstrated that dietary carbohydrate intake among Saudi T2DM patients is high, leading to poor glycaemic control [29].

Diabetic patients find it difficult to control portions while they strive to maintain balanced macronutrient levels, and they aim to decrease their consumption of desserts and sweet beverages. Research studies in the GCC region demonstrate that patients have a good awareness of dietary necessities yet struggle to maintain them daily, particularly during occasions such as Ramadan and Eid festivals [32,33]. Studies have shown that eating behaviour when under stress presents another obstacle, according to the available studies. According to Al-Ozairi A. et al. from Kuwait, their participants with depression had problems with general eating plans and carbohydrate intake [34]. Similarly, Fayed A. et al. from Saudi Arabia stated that diabetes-related distress is a significant challenge in diabetes management, including dietary adherence [35].

#### 3.1.3. Physical Activity Level

Regular physical activity is critical for enhancing insulin sensitivity and controlling blood glucose levels. Evidence shows that inactivity continues to affect many T2DM patients across the GCC region. Extreme temperatures, lack of walking-friendly environments, and costly fitness facilities further limit physical activities in this region. Women experience particular social limitations that restrict their ability to exercise physically. Research indicates that female patients in the GCC region face cultural limitations that prevent them from exercising regularly [36,37]. For example, Alfetni A. et al. mentioned that female T2DM patients were more uncomfortable with performing physical activities in public areas [36]. Several patients consider everyday household chores a sufficient amount of physical exercise despite their inadequacy in meeting established guidelines [38].

#### 3.1.4. Self-Monitoring of Blood Glucose

Patient participation in self-monitoring of blood glucose (SMBG) allows them to understand how their lifestyle choices affect their blood sugar, thus improving diabetes management [39]. SMBG practices in the GCC region differ considerably because they depend on patients’ education level, combined with their economic status and access to healthcare opportunities [23,40,41]. A study by Jamal A et al. stated that SMBG is more than 90% adequate among Saudi study participants. Furthermore, their research observed that the younger participants’ trend is toward digital monitoring of their blood sugar [40]. However, other studies in the GCC region have shown variable self-monitoring behaviours. For example, AlRasheed AY et al. stated that only about 14% of T2DM patients adhered to high levels of SMBG [42]. The variations across the studies could be due to the availability of healthcare services across the region. SMBG monitoring functions as an obligation to fulfill provider needs for some patients instead of serving their self-care needs.

#### 3.1.5. Other Self-Management Components

In addition to medication, diet, exercise, and glucose monitoring, comprehensive diabetes self-management includes coping with psychological stress, preventive care practices, and risk reduction behaviours. Diabetes management is influenced substantially by the emotional state of patients. Depression, along with diabetes distress and anxiety, frequently affects individuals with type 2 diabetes in GCC countries, yet healthcare providers frequently fail to detect these problems [43,44]. For example, a study by Al-Ozairi et al. in Kuwait stated that self-management practices were poor among T2DM patients with depression [34]. People experiencing psychological distress tend to perform worse in self-care by using less medication and making unhealthy dietary choices. The same scenario was observed in other GCC nations, such as the UAE [45], Saudi Arabia [44], and Oman [46]. Few people can access mental health care, and negative social views toward seeking support for mental health act as an additional challenge [47,48,49]. However, most of the studies are cross-sectional. Hence, the lack of causal and temporal associations limits further understanding.

Additionally, the use of traditional and complementary medicine among T2DM patients also plays a role in self-management behaviour and glycemic control. In a cross-sectional study from Taif (Saudi Arabia), 33.7% of diabetes patients reported using complementary/traditional medicine alongside standard care. Notably, 87% of the users had not consulted their doctor prior to use, and many used multiple remedies from multiple information sources [50]. A study from the UAE stated that among the users of complementary and traditional medicine, only about one in four users informed their physician about using traditional therapies [51].

Patient success in the prevention of complications requires regular foot examinations, eye assessments, and routine monitoring of cardiovascular health status. Patient participation in preventive screening varies throughout the entire region [52,53,54,55]. Many patients lack awareness of the importance of routine foot checks or assume that the absence of symptoms indicates no need for preventive care. A recent study in 2024 by Alkalash SH et al. in the GCC region stated that the majority of patients had inadequate knowledge and did not care for their foot-related problems [56].

In summary, T2DM patients who manage diabetes across GCC nations follow a wide range of practices with their self-care, which results from numerous social and healthcare system factors. Culturally sensitive healthcare solutions, system-wide interventions addressing individual and structural barriers, will be necessary to maintain patient practice improvements in diabetes self-management.

### 3.2. Patient Perspectives, Determinants, and Barriers of Self-Management

The effectiveness of T2DM self-management depends largely on patients’ perspectives, motivational aspects, barriers encountered, and actual world responsibilities. Unique cultural and social components and healthcare settings form the determinants of these factors within the GCC countries. The development of effective self-management and long-term health outcomes requires knowledge of patient perspectives alongside a complete understanding of the determinants of self-management, as this information helps in the design of improvement strategies.

#### 3.2.1. Knowledge and Understanding of Diabetes

Patients’ knowledge and understanding of diabetes vary widely across the GCC region [57,58,59]. Patients demonstrate that adequate knowledge of diabetes is a chronic issue; however, most people lack a detailed understanding of diabetes complications alongside their supervision targets and self-care expectations. Patients who have basic knowledge that diabetes needs medications along with dietary modifications often fail to comprehend key concepts regarding appropriate blood glucose targets, the importance of regular blood sugar monitoring, and long-term complications [59,60,61,62]. A recent study in Saudi Arabia demonstrated a significant knowledge gap among T2DM patients regarding several domains of T2DM, including diet and medication adherence. They also observed a positive correlation between diabetes knowledge and medication adherence, a component of self-management behaviour [60]. Furthermore, Alsaleh FM et al. from Kuwait suggested that healthcare providers play a significant role in improving patients’ knowledge and self-care practices, mainly in foot care [52]. A significant proportion of patients in studies conducted in the GCC region failed to show a proper understanding of HbA1c target ranges and hypoglycaemic symptom recognition [61,63,64]. Patient self-care practices improve when people with diabetes have better knowledge of disease management; therefore, education remains crucial for successful diabetes management.

#### 3.2.2. Attitudes and Beliefs Toward Self-Management

Some authors have evaluated the attitudes of T2DM patients toward different components of self-care practices, such as foot care, Insulin therapy, etc. [65,66,67]. Healthcare outcomes improve when patients link self-management responsibilities to personal control over their condition because they better adhere to medical treatment and wellness routines. People who carry negative attitudes toward their disease progression or question the value of treatments tend to neglect their self-care responsibilities. Some domains depicted a positive attitude leading to good practice, whereas some domains had inadequate responses [67,68,69]. The attitudes of GCC patients frequently follow cultural and religious norms in their region. Religious practices, which include Ramadan fasting, pose unique challenges for patients. Numerous patients choose to abstain from food despite doctors’ warnings about complications that arise when fasting without proper preparation. Healthcare providers must handle attitudinal obstacles with care because patients’ behaviours stem from their cultural and religious beliefs [33,70,71].

#### 3.2.3. Motivators and Barriers to Effective Self-Management

A range of elements motivates patients to take an active part in diabetic self-management practices. In GCC countries, family support is a leading factor in promoting self-management. Patients benefit from family involvement since families act as reminders for medication use, assist with dietary procedures and physical activity encouragement, and offer psychological support. For example, Alrasasimah, W. A. and Alsabaani A. et al. 2024 stated that perceived social support is one of the positive predictors of self-management practices among T2DM patients [72]. To support this, Kerari A suggested implementing personal and family support models to improve self-management among T2DM patients [73].

Several barriers exist that block effective self-management practices among patients with T2DM in the GCC region, even though positive factors exist. For example, a study performed in Qatar by Hassan A. et al. found several barriers, including work-related stress, work hours, and testing costs [74]. Some patients now participate more actively in disease monitoring through mobile apps and telehealth services because of increased digital health tool availability. Some barriers exist not only for traditional care but also for recent advances, including mHealth and artificial intelligence, as discussed by several authors [75,76,77]. Self-management in T2DM patients faces numerous psychological barriers that remain unrecognized, although they commonly appear together with depression, anxiety, and diabetes-related distress [43,44]. The poor financial status of both expats and lower-income families creates economic barriers to accessing glucometers, test strips, and insurance-uncovered medications. Unequal access to diabetes educators, disrupted care between different services, and insufficient educational resources tailored to specific cultures constitute health system barriers to prolonged self-care management [78,79,80]. Patients’ self-management experiences and interventions are heavily affected by their sociodemographic details, including their age, gender, education level, financial status, and geographical location [41,72,81]. Elderly patients commonly have multiple health conditions and body limitations, which create challenges for both physical activities and complex treatment protocols. The existing gender norms restrict women’s exercise opportunities because these social environments provide limited access to facilities for public exercise. Our statements are supported by Alsayed Hassan D et al. from Qatar. In their study, females had significantly lower levels of diabetes self-management practices than their male counterparts [74]. Being married and patients with a longer duration of diabetes are also important predictors for poor self-management behaviour, as stated by Al-Qahtani AM [82]. Age, income, and healthcare accessibility were other significant factors related to self-management behaviours among T2DM patients. Nonetheless, these sociodemographic factors vary across the studies from the GCC region [37,83,84]. This indicates that equitable and targeted intervention development needs to recognize sociodemographic inequalities to achieve effective healthcare delivery for all patient groups in the GCC region. In conclusion, T2DM patients’ perspectives, determinants, and barriers to self-management present a multidimensional concept in GCC countries. Strategic healthcare decisions implement these patients’ understanding to help T2DM patients achieve successful self-management practices that yield better healthcare results.

### 3.3. Knowledge Gaps

Multiple important knowledge gaps exist in self-management practices for T2DM despite the substantial advancements achieved in understanding and supporting this condition in GCC countries. Addressing these knowledge gaps becomes essential for developing better diabetes management approaches that are personalized and culturally appropriate throughout GCC countries. The defined areas with presently limited evidence serve as the foundation for future guidance aimed at shaping clinical practice together with research and policy decisions.

i.Limited research on diverse population subgroups: One of the critical gaps identified in this review is a significant lack of evidence and insufficient representation of diverse patient subgroups. Most research has concentrated on adults at urban tertiary care facilities while neglecting rural citizens, lower-income groups, and foreign nationals. Notably, the GCC region has significant demographic variations, including among expatriate communities.ii.Fragmented assessment of psychosocial and mental health factors: Some authors have attempted to find the relationship between mental health well-being and self-management practices among T2DM patients. However, these studies are limited, and there is a lack of comprehensive assessment.iii.Insufficient evaluation of cultural and religious influences: the impact of cultural and religious elements on diabetes self-management in GCC countries has been recognized, but systematic investigations into their effects are lacking.iv.Gaps in technology adoption and digital health interventions: The vast use of smartphones and internet connectivity in GCC countries creates new opportunities to support diabetes self-management through digital health tools. Studies evaluating the effectiveness, sustainability, and usability of mobile health applications and telemedicine services, wearable glucose monitors, and remote coaching programs are scarce in the GCC region. The majority of current research shows only brief outcomes from pilot tests with weak outcome evaluation methods.v.Underexplored the role of healthcare system factors: The majority of studies have focused on patient-oriented factors and their association with self-management behaviour. Studies that explore factors related to health systems, including the availability and accessibility of multidisciplinary teams, diabetes educators, continuity care, and referral systems, are limited.vi.Lack of higher levels of evidence from longitudinal and interventional research designs: most of the existing evidence and conclusions related to self-management behaviour are based on cross-sectional studies, limiting policymakers from drawing strong inferences for implementing necessary changes.vii.Limited focus on patient-centred outcomes: Research conducted in the GCC region has focused mostly on clinical outcomes such as glycaemic control, complications, etc, with self-management behaviour. However, research studies that explored patient-oriented outcomes through qualitative analysis are limited.

## 4. Discussion

### 4.1. Interpretative Commentary and Implications

The present review findings highlight that, beyond access to medicines, empowering patients through sustained education and literacy programs is a critical system-level strategy in the region. Health literacy is positively correlated with medication adherence. Studies in the GCC region indicate that tailored health literacy programs with ongoing evaluation and continuous improvement enhance patient outcomes and adherence [85,86,87,88]. A recent study in 2023 stated that adding mobile phone text-message reminders to usual care significantly improved medication adherence in T2DM patients [89]. A real-world study reported that patients who used a diabetes smartphone app (with data tracking and feedback) had greater reductions in blood glucose, highlighting how digital tools plus regular self-monitoring enhance adherence and glycemic control [90]. Furthermore, the combination of counselling with medication reminders, patient education services in programs, mHealth, and digital solutions can improve treatment adherence [91,92]. This suggests that integrating traditional counselling with innovative digital health tools may offer a sustainable pathway to improve adherence in diverse GCC populations.

Health promotion activities to improve eating habits include diabetes-friendly cooking courses alongside modified dietary guidelines and one-on-one dietary education (personalized) provided by dietitians. The integration of culturally acceptable religious ceremonies and traditional food choices in therapeutic counselling produces better patient responses to dietary plans [79,93,94]. A UAE cluster-RCT of a culturally adapted 12-month lifestyle program where intervention patients (receiving diet counseling aligned with local food preferences and Ramadan considerations) showed significantly greater reductions in caloric intake and weight than controls did, demonstrating better dietary compliance with culturally tailored guidance [95]. An expert consensus (“Transcultural Diabetes Nutrition Algorithm—Middle Eastern Version”), which emphasizes that dietary advice tailored to regional foods and cultural practices (e.g., accommodating traditional cuisines and Ramadan fasting) greatly improves patient adherence to nutritional plans [96].

Religious and cultural practices are very important determinants of dietary and self-management practices in the GCC countries. Changes in meal timing during Ramadan and extension of the fasting periods have significant impacts on medication compliance, dietary consumption, and blood sugar levels. Culturally sensitive pre-Ramadan and during-Ramadan counselling has led to improvements in self-administration and decreases in complications [97]. Similarly, social culture related to eating and hospitality may promote the intake of high-calorie foods, which can detract from dietary counseling unless there is the active participation of the family and community [95]. Positive family support has been linked with greater treatment adherence (to diet and medications), whereas the absence of such support can inhibit lifestyle changes. This is an important reason why culturally sensitive, family-inclusive, religiously adapted programs are essential in the context of diabetes self-management throughout the GCC region [98]. These culturally tailored approaches illustrate that interventions are more effective when they respect the traditional food practices and social customs of GCC patients. However, healthcare access to experienced nutrition experts exists only within big urban areas, and in most places, it is a big challenge to implement these implications.

This review identified that in the GCC context, extreme climate conditions, limited walkable environments, costly fitness facilities, and particularly cultural barriers for women restrict opportunities for regular exercise. To overcome this limitation, home-based physical activity programs or other suitable environments could be beneficial [38,99]. Few GCC nations have launched programs promoting indoor walking at shopping malls, personalized home exercises, and culturally suitable fitness programs. These initiatives, such as the Step into Health (SIH) program in Qatar, demonstrate initial achievements in promoting activity, yet poor long-term adherence occurs because of the lack of continuous motivation, sociocultural limitations, and structured follow-up procedures [100,101].

Education sessions teach diabetes patients about glucose pattern analysis, alongside individual provider assessments, and serve as key strategies to enhance SMBG practices. Smartphone applications and digital tools connected to glucose monitors function as emerging technology-based solutions for enhancing the use of SMBG data by tech-savvy populations [102,103,104]. These strategies show that leveraging education and digital innovations together can improve patient engagement and enable more proactive disease management. These studies’ findings suggest that educational programs, together with structured screening models, can enhance preventive care engagement [53,55]. Nevertheless, several obstacles, including low awareness levels, poor practices, and the restrictiveness of system structures, continue to affect the GCC region. Proper diabetes complication prevention requires specific interventions and policy changes to address these ongoing problems. This reflects the broader need for healthcare systems to combine awareness campaigns with practical service delivery models that ensure equitable access to preventive care.

The motivation to actively engage in diabetes self-management strongly depends on positive and trusting relationships with healthcare providers. Patients whose physicians create feelings of respect combined with active listening and guidance tend to follow advice regarding recommended practices. Therefore, healthcare providers could be essential facilitators for improving self-management practices among T2DM patients, as suggested by studies in the GCC region [78,94]. These observations emphasize the central role of healthcare providers not only as clinical managers but also as facilitators of patient empowerment.

Furthermore, social and cultural elements hinder patients’ abilities to effectively manage T2DM in the GCC region. Local eating traditions featuring carbohydrate-heavy and sweet offerings combined with cultural customs of generous eating habits and social roles define gender standards, which intersect to create obstacles for patients with T2DM trying to maintain healthy diets and workout routines [29,93,105]. Multiple barriers illustrate the necessity for health services to implement complete interventions that address psychological needs and social factors alongside economic constraints and systemic institutional constraints.

### 4.2. Future Directions

Future research integrating more inclusive strategic actions for comprehensive diabetes self-management is required in the GCC region. Research must concentrate on studying the experiences of varied and low-represented populations, including rural inhabitants, expatriates, and persons with multiple medical conditions, to guarantee intervention equality and complete representation. Healthcare professionals should perform psychological and mental health assessments during routine diabetes care because emotional health directly affects how patients handle their condition. Interventions and educational programs related to diabetes care should be developed with cultural sensitivity to consider religious customs and family dynamics so that patients can accept these programs more easily.

Digital health technology penetration creates new self-management support options, but thorough assessments of these platforms are needed to prove their value and usability across patient demographics. The implementation of standardized regionally modified diabetes self-management education curricula throughout GCC countries will create clear patient teaching approaches while driving best practices. Healthcare institutions must build stronger interdisciplinary models that deliver continuous patient-centred support for self-management programs beyond single clinical meetings. Longitudinal investigations and interventional studies must become the focus of future research because they enable better observation of behavioural transformations in addition to sustained impact on clinical results and patient-oriented measurements, including quality of life, personal confidence, and care satisfaction. Such GCC countries must coordinate research studies with healthcare delivery improvements and policy initiatives to intensify their patient empowerment strategies, which lead to better, sustainable diabetes control practices and improved patient outcomes.

### 4.3. Limitations

Even though the present review was conducted with standard methods, some limitations should be acknowledged. First, as a narrative review, we did not apply a formal quality appraisal tool (e.g., the Newcastle–Ottawa Scale or JBI checklist), which is typically mandatory for systematic reviews. Instead, we sought to maintain rigor through predefined inclusion/exclusion criteria, independent double screening, and thematic synthesis. Second, most of the included studies were cross-sectional in design, which restricts the ability to infer causality, introduces potential self-reporting bias, and limits the generalizability of findings. These methodological constraints should be considered when interpreting the results. Nevertheless, the review provides valuable insights by collating and thematically integrating the most recent evidence on self-management behaviours in GCC populations.

## 5. Conclusions

This review provides the first updated synthesis (2020–2025) of self-management behaviours in T2DM patients across all GCC countries, with a particular focus on the cultural, systemic, and socioeconomic factors that shape patient practices. Our review revealed that despite advances in healthcare infrastructure and increased awareness, substantial gaps persist in the self-management behaviours of patients living with diabetes. We also highlighted several determinants of self-management behaviours, such as cultural, mental health, and socioeconomic factors. Therefore, it is critical to develop culturally sensitive patient-centred care, individualized educational programs, and adopt supportive digital solutions to enhance diabetes-related self-care management. The findings of the present review demonstrate the pressing need for ongoing policy-driven interventions to bridge these gaps and improve long-term prognosis in T2DM patients in the GCC region. Furthermore, longitudinal investigations and interventional studies must become the focus of future research because they enable better observation of behavioural transformations in addition to sustained impact assessments of clinical results and patient-oriented outcomes.

## Figures and Tables

**Figure 1 healthcare-13-02247-f001:**
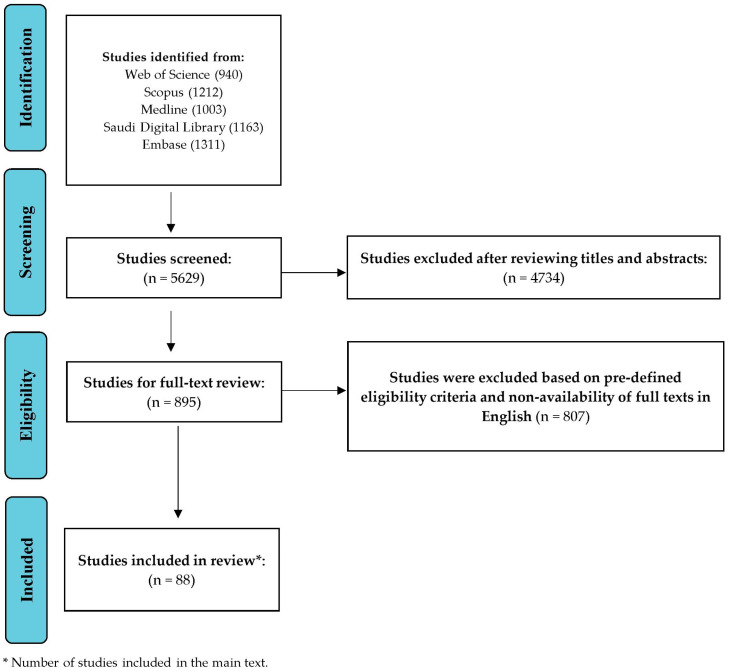
Study screening and selection process.

**Table 1 healthcare-13-02247-t001:** Search strategies summary.

Component	Details
Databases Searched	Web of Science, Scopus, Medline, Saudi Digital Library, Embase
Rationale for Database Selection	Selected for accessibility, coverage, and relevance to T2DM self-management in GCC countries
Search Period	January 2020–March 2025
Search Terms/Keywords	(“type 2 diabetes” OR T2DM) AND (“self-management” OR “self-care”) AND (barriers OR facilitators) AND (Saudi Arabia OR Qatar OR UAE OR Kuwait OR Oman OR Bahrain)
Boolean Operators Used	“AND”, “OR”, “NOT”
Inclusion Criteria	Studies on adult (≥18 years) T2DM patients, peer-reviewed, English language, with DOI, published between January 2020 and March 2025
Exclusion Criteria	Studies on type 1 or gestational diabetes, grey literature, unpublished works and studies not focused on T2DM self-management
Screening Method	Two independent reviewers screened titles and abstracts; discrepancies were resolved with a third reviewer
Number of Articles Included	80 articles

## Data Availability

This study is a narrative review and does not involve the generation or analysis of new data.

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
