# Peer review of "Self-Management Behaviours in Type 2 Diabetes Across Gulf Cooperation Council Countries: An Updated Narrative Review to Enhance Patient Care"

_healthcare, 2025, doi:10.3390/healthcare13172247_

Round 1
Reviewer 1 Report
Comments and Suggestions for Authors
The topic addresses a significant public health concern with a high prevalence—type 2 diabetes mellitus (T2DM)—and focuses on patient self-management in the Gulf Cooperation Council (GCC) countries. The emphasis on cultural and systemic factors influencing patient behavior represents a valuable contribution to understanding region-specific challenges and supports the development of targeted interventions.
Introduction: The introduction is appropriately structured and clearly outlines the epidemiological background of T2DM in the GCC region. The authors effectively present the rationale for conducting this review and successfully identify the existing research gap. However, some content is repeated, which affects the precision and readability of the section. I recommend condensing the text and formulating a more concise and explicit statement of the review’s objective(s) and guiding research questions.
Main Findings and Discussion: I suggest separating the current combined section into two distinct parts: 3. Results and 4. Discussion. This structural revision would improve the logical flow, enhance clarity, and help readers differentiate between synthesized empirical findings and interpretative commentary. Please also verify internal consistency between both sections and reassign overlapping content as appropriate. Additionally, I strongly recommend incorporating a tabular summary of key studies (e.g., country, year, study design, and main findings) into the Results section. This would provide readers with a clearer and more accessible overview of the included evidence and enhance the methodological transparency of the review.
Knowledge Gaps and Future Directions: I recommend integrating the Knowledge Gaps subsection into the concluding part of the Results section and relocating the Future Directions subsection to the final portion of the Discussion. This restructuring would improve narrative coherence, eliminate redundancy, and align the manuscript with standard conventions in narrative reviews.
Conclusions: The conclusion would benefit from further refinement to reinforce the unique contribution of this review.
Author Response
"Please see the attachment."

General:
The authors would like to thank the reviewer for the precious time spent reviewing the paper and his excellent suggestions for improving it. Efforts have been made to modify the paper as per the reviewer’s suggestions and recommendations. The authors will be happy to hear a positive reply. All the points included according to the reviewer’s comments can be seen in track changes.
Specific response to the reviewer’s suggestions:
Kindly find the attached response to each question one by one:
Point 1: The topic addresses a significant public health concern with a high prevalence—type 2 diabetes mellitus (T2DM)—and focuses on patient self-management in the Gulf Cooperation Council (GCC) countries. The emphasis on cultural and systemic factors influencing patient behavior represents a valuable contribution to understanding region-specific challenges and supports the development of targeted interventions.
Response 1: Thanks for the comment. The authors are pleased to hear the reviewer for recognizing the importance of our chosen topic and the emphasis on cultural and systemic factors. We appreciate that the reviewer found our focus on region-specific challenges to be a valuable contribution, as one of our primary objectives was to highlight the unique determinants of diabetes self-management in the GCC region to guide more targeted interventions.
Point 2: The introduction is appropriately structured and clearly outlines the epidemiological background of T2DM in the GCC region. The authors effectively present the rationale for conducting this review and successfully identify the existing research gap. However, some content is repeated, which affects the precision and readability of the section. I recommend condensing the text and formulating a more concise and explicit statement of the review’s objective(s) and guiding research questions.
Response 2: Thanks for the comment. The authors are pleased to hear the reviewer’s positive comments on the introduction and rationale. According to the reviewer’s suggestions, we deleted some repeated sentences from the introduction.
Point 3: I suggest separating the current combined section into two distinct parts: 3. Results and 4. Discussion. This structural revision would improve the logical flow, enhance clarity, and help readers differentiate between synthesized empirical findings and interpretative commentary. Please also verify internal consistency between both sections and reassign overlapping content as appropriate.
Response 3: Thanks for the comment. According to the reviewer’s suggestions, we have revised the manuscript structure by separating the combined “Main Findings and Discussion” into two distinct sections: “Results” and “Discussion.” Within the Discussion, we introduced a new subsection “4.1. Interpretative Commentary and Implications” to provide context and synthesis with existing literature. The “Knowledge Gaps” subsection has been moved under the Results (3.3), while “Future Directions” has been relocated to the concluding part of the Discussion (4.2). The authors believe the restructuring (according to the reviewer’s comments) improved improved the logical flow, enhance clarity, and help readers differentiate between synthesized empirical findings and interpretative commentary.
Point 4: Additionally, I strongly recommend incorporating a tabular summary of key studies (e.g., country, year, study design, and main findings) into the Results section. This would provide readers with a clearer and more accessible overview of the included evidence and enhance the methodological transparency of the review.
Response 4: Thanks for the comment. According to the reviewer’s comments, the authors included a new table (Table 2) that summarizes the key studies.
Point 5: Knowledge Gaps and Future Directions: I recommend integrating the Knowledge Gaps subsection into the concluding part of the Results section and relocating the Future Directions subsection to the final portion of the Discussion. This restructuring would improve narrative coherence, eliminate redundancy, and align the manuscript with standard conventions in narrative reviews.
Response 5: Thanks for the comment. Please find the “Response 3”.
Point 6: Conclusions: The conclusion would benefit from further refinement to reinforce the unique contribution of this review.
Response 6: Thanks for the comment. The authors made the changes as per the reviewer’s suggestions.
The authors thank the reviewer once again for the positive and constructive comments.
Reviewer 2 Report
Comments and Suggestions for Authors
The review includes only peer-reviewed studies published between January 2020 and March 2025. This may exclude relevant research published outside this timeframe, potentially leading to an incomplete picture.The review doesn't mention a formal quality assessment of included studies, which could affect the reliability of conclusions. And also, you have not explicitly described how you evaluated or tested your approach in analysing the studies.
I recommend, if possible you make these minor corrections which will strengthen your article.
Author Response
"Please see the attachment."

General:
The authors would like to thank the reviewer for the precious time spent reviewing the paper and his excellent suggestions for improving it. Efforts have been made to modify the paper as per the reviewer’s suggestions and recommendations. The authors will be happy to hear a positive reply. All the points included according to the reviewer’s comments can be seen in track changes.
Specific response to the reviewer’s suggestions:
Kindly find the attached response to each question one by one:
Point 1: The review includes only peer-reviewed studies published between January 2020 and March 2025. This may exclude relevant research published outside this timeframe, potentially leading to an incomplete picture.
Response 1: Thanks for the comment. We limited our review to studies published between January 2020 and March 2025 to ensure that the synthesis reflects the most recent evidence on self-management practices of T2DM patients in the GCC region. The authors selected this time frame to get the most updated evidence as self-management strategies underwent notable changes following the COVID-19 pandemic (post-pandemic). The rationale for this is included in the revised manuscript.
Point 2: The review doesn't mention a formal quality assessment of included studies, which could affect the reliability of conclusions. And also, you have not explicitly described how you evaluated or tested your approach in analysing the studies.
Response 2: Thanks for the comment. The present study is a narrative review, and to maintain the quality, transparency, and reliability of the included studies, the authors used several methods (please find the summary in Table 1), including pre-defined criteria. However, the authors did not apply a quality appraisal tool (such as Newcastle-Ottawa Scale, JBI critical appraisal, etc), which is mandatory in a Systematic review.
Point 3: I recommend, if possible you make these minor corrections which will strengthen your article.
Response 3: Thanks for the comment. We have included all suggested details in the revised manuscript according to the reviewer's suggestions.
The authors thank the reviewer once again for the positive and constructive comments.
Reviewer 3 Report
Comments and Suggestions for Authors
This manuscript presents a narrative review of self-management behaviors in patients with Type 2 Diabetes Mellitus (T2DM) across Gulf Cooperation Council (GCC) countries. The authors have made a significant effort in compiling data from various sources and organizing it under thematic subtopics such as medication adherence, dietary habits, physical activity, self-monitoring, and socio-cultural determinants. The subject is of high public health relevance, and the regional focus adds contextual specificity that is often missing in global reviews. However, the manuscript has several conceptual, methodological, and structural shortcomings that must be addressed to meet the standards of a peer-reviewed healthcare journal. There is no comparative or evaluative angle—for instance, which interventions are most effective, or which behavioral factors are most modifiable? The topic is important and timely, but the manuscript falls short in offering a clear conceptual framework. The paper reads more like an annotated summary of studies rather than a critical synthesis or integrative analysis. Although described as a "narrative review," the methodology section attempts to follow a quasi-systematic approach (e.g., multiple databases, inclusion/exclusion criteria). However, it lacks PRISMA flow details, quality assessment of included studies, or evidence grading. The discussion mostly summarizes individual studies without conducting thematic integration, identifying patterns, or distinguishing high- vs. low-quality evidence. There is an overreliance on cross-sectional studies, yet no meaningful discussion of their limitations. While the manuscript refers to Ramadan fasting, communal eating, and family support, these factors are discussed superficially. More structured analysis of religion/culture as both facilitators and barriers is warranted. There’s no exploration of how traditional medicine (e.g., herbal use) intersects with or undermines self-management practices. The manuscript has potential to contribute significantly to regional diabetes care literature. However, it requires a deeper analytical approach, clearer structure, improved methodological transparency, and editorial refinement to meet publication standards.
Comments on the Quality of English Language
The manuscript needs extensive language editing to correct grammatical issues, redundancy, and syntax errors.
Author Response
Please see the attachment

General:
The authors would like to thank the reviewer for the precious time spent reviewing the paper and his excellent suggestions for improving it. Efforts have been made to modify the paper as per the reviewer’s suggestions and recommendations. The authors will be happy to hear a positive reply. All the points included according to the reviewer’s comments can be seen in track changes.
Specific response to the reviewer’s suggestions:
Kindly find the attached response to each question one by one:
Point 1: This manuscript presents a narrative review of self-management behaviors in patients with Type 2 Diabetes Mellitus (T2DM) across Gulf Cooperation Council (GCC) countries. The authors have made a significant effort in compiling data from various sources and organizing it under thematic subtopics such as medication adherence, dietary habits, physical activity, self-monitoring, and socio-cultural determinants. The subject is of high public health relevance, and the regional focus adds contextual specificity that is often missing in global reviews. However, the manuscript has several conceptual, methodological, and structural shortcomings that must be addressed to meet the standards of a peer-reviewed healthcare journal.
Response 1: Thanks for the positive comment on the topic. We are delighted to hear from the reviewer regarding the manuscript’s contributions and public health importance. We have carefully revised the manuscript in response to the reviewer’s detailed comments, clarifying methodological choices, refining the conceptual framing, and restructuring sections where appropriate. These revisions, explained point by point below, aim to strengthen both the rigor and clarity of the paper
Point 2: There is no comparative or evaluative angle—for instance, which interventions are most effective, or which behavioral factors are most modifiable?
Response 2: Thanks for the comment. The authors agree that a comparative and evaluative perspective enhances the utility of a narrative review. In the revised manuscript, we restructured the main text (Main findings and Discussion) into two different sections. In the revised manuscript, we have revised the Discussion section (4.1) to highlight, within each thematic area, which interventions appear most effective and which behavioral factors are more modifiable in the GCC context.
Point 3: The paper reads more like an annotated summary of studies rather than a critical synthesis or integrative analysis. Although described as a "narrative review," the methodology section attempts to follow a quasi-systematic approach (e.g., multiple databases, inclusion/exclusion criteria). However, it lacks PRISMA flow details, quality assessment of included studies, or evidence grading. The discussion mostly summarizes individual studies without conducting thematic integration, identifying patterns, or distinguishing high- vs. low-quality evidence.
Response 3: Thanks for the comment. Thanks for the comment. The present study is a narrative review, and to maintain the quality, transparency, and reliability of the included studies, the authors used several methods (please find the summary in Table 1), including pre-defined criteria. However, the authors did not apply a quality appraisal tool (such as Newcastle-Ottawa Scale, JBI critical appraisal, etc), which is mandatory in a Systematic review. Furthermore, the new discussion section consists of subsection (4.2) that discusses the suggestions given by the reviewer.
Point 4: There is an overreliance on cross-sectional studies, yet no meaningful discussion of their limitations.
Response 4: Thanks for the comment. According to the reviewer’s suggestions, we included a separate section (4.3) for the limitations of the present review.
Point 5: While the manuscript refers to Ramadan fasting, communal eating, and family support, these factors are discussed superficially.
Response 5: Thanks for the comment. We agree that cultural and religious factors such as Ramadan fasting, communal eating, and family support required deeper elaboration. Accordingly, we have expanded section 4.1 to critically discuss how these factors influence self-management, and we added supporting references
Point 6: There’s no exploration of how traditional medicine (e.g., herbal use) intersects with or undermines self-management practices.
Response 6: Thanks for the comment. According to the reviewer’s suggestions, we explored and included details related to traditional medicine in the revised manuscript (Both results and discussion section).
Point 7: The manuscript has potential to contribute significantly to regional diabetes care literature. However, it requires a deeper analytical approach, clearer structure, improved methodological transparency, and editorial refinement to meet publication standards.
Response 7: Thanks for the comment. The authors believe that the reviewer’s comments are acknowledge and corrected in the revised manuscript. Furthermore, we are ready for any further comments from the reviewer.
The authors thank the reviewer again for the positive and constructive comments.